



# Altitude profiles of CCN characteristics across the Indo-Gangetic Plain prior to the onset of the Indian summer monsoon

Venugopalan Nair Jayachandran[1], Surendran Nair Suresh Babu[1*], Aditya Vaishya[2], Mukunda M. Gogoi[1], Vijayakumar S Nair[1], Sreedharan Krishnakumari Satheesh[3,4], Krishnaswamy Krishna Moorthy[4]

[1] Space Physics Laboratory, Vikram Sarabhai Space Centre, ISRO PO, Thiruvananthapuram, India.

[2] School of Arts and Sciences, Ahmedabad University, Ahmedabad, India.

[3] Centre for Atmospheric and Oceanic Sciences, Indian Institute of Science, Bangalore, India.

[4] Divecha Centre for Climate Change, Indian Institute of Science, Bangalore, India.

[*]Correspondence to: Surendran Nair Suresh Babu (sureshsplvssc@gmail.com)



**Abstract**

Concurrent measurements of the altitude profiles of cloud condensation nuclei (CCN) concentration, as a function of supersaturation (ranging from 0.2% to 1.0%), and aerosol optical properties (scattering and absorption coefficients) were carried out aboard an instrumented aircraft across the Indo-Gangetic Plain (IGP) covering coastal, urban and arid environments, just prior to

the onset of the Indian summer monsoon (ISM) of 2016, under the aegis of the SWAAMI - RAWEX campaign. In general, the CCN concentration has been highest in the Central IGP, decreasing spatially from east to west above the planetary boundary layer (PBL), which is ~1.5 km for the IGP during pre-monsoon. Despite of this, the CCN activation efficiency at 0.4% supersaturation has been, interestingly, the highest over the eastern IGP (~72%), followed by the

west (~61%), and has been the least over the central IGP (~24%) within the PBL. In general, higher activation efficiency is noticed above the PBL than below it. The Central IGP showed remarkably low CCN activation efficiency at all the heights, which appears to be associated with high black carbon (BC) mass concentration there, indicating the role of anthropogenic sources in suppressing the CCN efficiency. First ever CCN measurements over the western IGP, encompassing 'The

Great Indian desert', show high CCN efficiency, ~61% at 0.4% supersaturation, indicating hygroscopic nature of the dust. The vertical structure of CCN properties is found to be airmass-dependent; with higher activation efficiency even over the central IGP during the prevalence of marine airmass. Precipitation episodes seem to reduce the CCN activation efficiency below cloud level. An empirical relation has emerged between the CCN concentration and the scattering aerosol

index (AI), which would facilitate prediction of CCN from aerosol optical properties.


## 1.    Introduction

Cloud nucleating ability of aerosols is fundamental in understanding the aerosol-cloud interactions (ACI) and associated feedback processes, which are complex in nature and pose a major challenge in quantifying the indirect climate forcing of aerosols (*Boucher et al., 2013; IPCC 2013*). Cloud Condensation Nuclei (CCN) form a sub-set of atmospheric aerosols (also known as Condensation Nuclei, CN) and take part in cloud processes, accelerate the condensation of water vapour leading to the formation of liquid cloud droplets and modify the microphysical properties of clouds depending on the number size distribution, chemical composition, and mixing state of aerosols (*Dusek et al., 2006; Farmer et al., 2015; Zhang et al., 2017*). Several investigators have examined temporal and spatial distribution of the CCN properties and their processing by non-precipitating clouds over both continental and marine environments (*Hoppel et al., 1973, Hudson and Xie, 1999, Jurányi et al., 2011, Paramonov et al., 2015, Schmale et al., 2018*). Significant variability in the CCN activation efficiency has also been reported over regions influenced by urban (*Sotiropoulou et al., 2007*) and industrial emissions (*Asa-Awuku et al., 2011*). Efforts have also been made to infer or predict CCN properties based on aerosol concentration and optical properties (for example, *Jefferson, 2010; Liu and Li, 2014*). However, due to the region-specific and heterogeneous nature of the composition of aerosols, their chemical interactions, vertical mixing and advection to long distances, significant uncertainties still persist in characterizing the CCN activation efficiency, especially its region-specific nature and altitude variation in the realistic atmosphere (*Zhang et al., 2017*). The information on vertical distribution of the CCN number concentration, CCN efficiency and its variation with supersaturation are some of the vital parameters needed in quantifying the ACIs. In-situ measurements of the vertical distribution of the CCN activity especially over polluted regions are very important in accounting for the ACI in climate models (*Li et al., 2016*).

In the above context, the importance of South Asian region is unequivocal. Aerosol physicochemical properties show large spatio-temporal variation over this region owing to the



diverse source influence, both natural and anthropogenic, which show large seasonality and

dependence on large-scale meteorology (*Lawrence and Lelieveld, 2010; Babu et al., 2013*). Even

within South Asia, the Indo-Gangetic Plains (IGP) fall under those regions in the globe where very

high aerosol loading persists almost throughout the year *(Di Girolamo et al., 2004)* and also depict

a steady increasing trend in the Aerosol Optical Depth (AOD) *(Babu et al., 2013)*, increasing

surface dimming *(Padmakumari et al., 2007; Badrinath et al., 2010)*, and enhanced mid

tropospheric warming *(Satheesh et al., 2008)*. Through modelling efforts, *Vinoj et al., (2014)* have

shown possible linkages of West Asian dust loading over the Arabian Sea with the Indian summer

monsoon (ISM). The competing roles of natural (mostly mineral dust and marine aerosols) and

anthropogenic aerosols over this region and their high seasonality, aided by the large-scale

industrial and agricultural activities in this region and its particular orography makes the IGP one

of the best natural laboratory for investigating the complex aerosol impacts *(Moorthy et al., 2016)*.

Despite these, characterisation of the vertical structure and the spatial variability of the CCN

characteristics across the IGP remains quite limited, except for some recent efforts using

instrumented aircraft during the summer monsoon season under the Cloud Aerosol Interaction and

Precipitation Enhancement Experiment (CAIPEEX) (*Padmakumari et al., 2017, Konwar et al.,*

*2014, Prabha et al., 2012*). A few ground-based measurements also exist scattered across the sub-

continent (*Bhattu and Tripathy, 2014; Gogoi et al., 2015; Jayachandran et al., 2017; Singla et al.,*

*2017*).

In light of the above, and with a view to understand the ACI and its linkage to the ISM, an

experimental campaign was undertaken under the aegis of SWAAMI (South-West Asian Aerosol

- Monsoon Interactions) and RAWEX (Regional Aerosol Warming Experiment), executed jointly

by the Indian Space Research Organisation (ISRO) and the Ministry of Earth Sciences (MoES) of

India, and the Natural Environment Research Council (NERC) of the UK. Under this, concurrent

and collocated airborne measurements of the vertical structure of the CCN characteristics and





aerosol scattering and absorption coefficients were made across the IGP, just prior to the onset of

the ISM. The campaign details along with the measurement protocols are given below, followed

by the results and discussions.

## 2.    Experiment details, data and analysis

### 2.1. Campaign

Airborne measurements of the CCN number concentration as a function of supersaturation (0.2,

0.3, 0.4, 0.7, 1.0) along with the scattering and the absorption coefficients were carried out across

the IGP from 1$^{st}$ June till 20$^{th}$ June 2016, prior to onset of the ISM, using the instrumented research

aircraft of the National Remote Sensing Centre (NRSC) of ISRO. The details of the sorties, base

stations, instruments used etc. are listed in Table 1. Synoptic wind conditions, using ERA-Interim

reanalysis product from the European Centre for Medium-Range Weather Forecasts (ECMWF),

during June 2016, at two altitude levels (a) 975 hPa (near to the surface) and (b) 700 hPa (free-

tropospheric altitude), are shown in Figure 1. Near-surface advection of marine airmass is seen

over the peninsula and regions south of ~ 25°N, while to the north of it and at higher levels, dry

continental airmass is advected from the northwest. As per the India Meteorological Department

(IMD), onset of ISM during 2016 was on 8 June 2016 at Kerala coast which advanced to eastern

IGP by 10$^{th}$ June and reached central-IGP by 19$^{th}$ June.

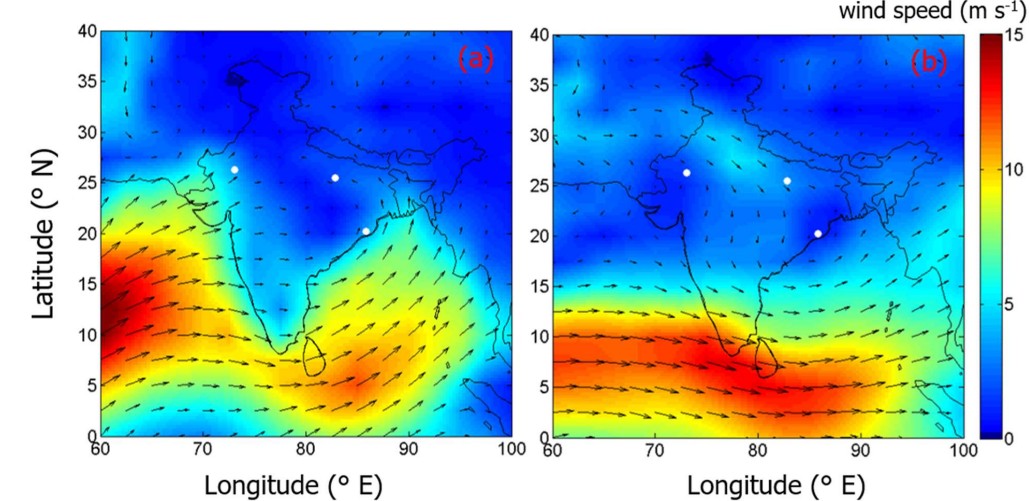

**Figure 1:** The strength and direction of winds at (a) 975 hPa and (b) 700 hPa over the Indian sub-continent during June 2016. White dots indicate the base stations. Wind data is from the ECMWF Era-Interim reanalysis.

Aircraft measurements were carried out from three base stations, each representing distinct regions of IGP viz. (i) Bhubaneswar (BBR, 20.24º N, 85.81º E, 42 m a.m.s.l.) – a semi-urban coastal location at eastern end of the IGP, (ii) Varanasi (VNS, 25.45º N, 82.85º E, 81 m a.m.s.l.) representing aerosol-laden (polluted) Central IGP, and (iii) Jodhpur (JDR, 26.25º N, 73.04º E, 219 m a.m.s.l.), representing semi-arid location on the western IGP, which receives large amount of mineral dust; lofted from the adjoining deserts as well as advected from the West Asian and East African regions. The base stations along with the direction of sorties for different days of campaign are shown in Figure 2a. The instruments aboard and the local weather conditions are listed in Table 1. As seen from the table, pre-monsoon showers occurred on two days at BBR and on one day at VNS. The campaign was executed just prior to onset of the ISM at each of the base station. The ISM started covering the IGP by around 19th of June 2016.

**Table 1:** Details of the sorties, including dates, instruments used, and, rain events for the campaign period.



| Region (Base Station) | Coordinates (°N, °E) | Height, m (a.m.s.l.) | Period (2016) | Remarks | Instruments |
|---|---|---|---|---|---|
| Eastern IGP (BBR) | 20.24, 85.81 | 42 | 1 - 5 June | Rain on 3[rd] and 4[th] June after the sorties | CCN counter (Model : CCN-100, Make : DMT) |
| Central IGP (VNS) | 25.45, 82.85 | 81 | 8 - 13 June | Rain on 7[th] June evening | CPC (Model: 3776, Make: TSI) Aethalometer (Model: AE-33, Make: Magee Scientific) |
| Western IGP (JDR) | 26.25, 73.04 | 219 | 17 - 20 June | No Rain | Nephelometer (Model: 3563, Make: TSI) |

All the aircraft sorties were carried out late in the forenoon to early afternoon (10 – 14 hours IST, IST standing for the Indian Standard Time, which is 05:30 hrs ahead of the UTC) to ensure that the planetary boundary layer (PBL) is fully evolved and aerosols are well mixed within the PBL. During this period, being summer over the Indian region, the PBL would be quite deep as the thermal convections would be strong providing a thorough vertical mixing. Mean PBL heights at

local noon time over the IGP regions, estimated from NCEP/NCAR global reanalysis product at 0.25° × 0.25° grid resolution data for the flight sortie days were found to be 1.4 ± 0.2 km, 2.3 ± 0.5 km, and 1.3 ± 0.5 km for BBR, VNS, and JDR, respectively (Vaishya et al., 2018). Due to unpressurised mode of operation of the aircraft, the ceiling altitude of airborne measurements was ~ 4 km amsl. In all, 14 sorties were made, 5 from each base station, except from JDR where only

4 sorties were made. Each sortie was for a period of ~ 3.5 hours, during which, the measurements were made at six altitude levels – ~ 500, 1000, 1500, 2000, 2500, and 3000 m above the ground level (a.g.l.), following the 'staircase pattern' shown in Figure 2b (*Babu et al., 2016*). Accordingly, after takeoff, the aircraft climbed to the first level (500 m a.g.l.), stabilized the attitude and flew at that level for ~ 30 minutes during which it covered a horizontal distance of ~ 150 km; before

climbing up to the next higher level and retracing the path. This procedure was repeated until the



5    highest level (ceiling altitude) was reached, after which the aircraft descended to the base. The

sorties were repeated on consecutive days, except that on each day the aircraft proceeded to a

different radial direction from the base, as shown in Figure 2a, so that the five sorties together

provided a gross picture of the aerosol properties around the base station within a radius of about

150 km.

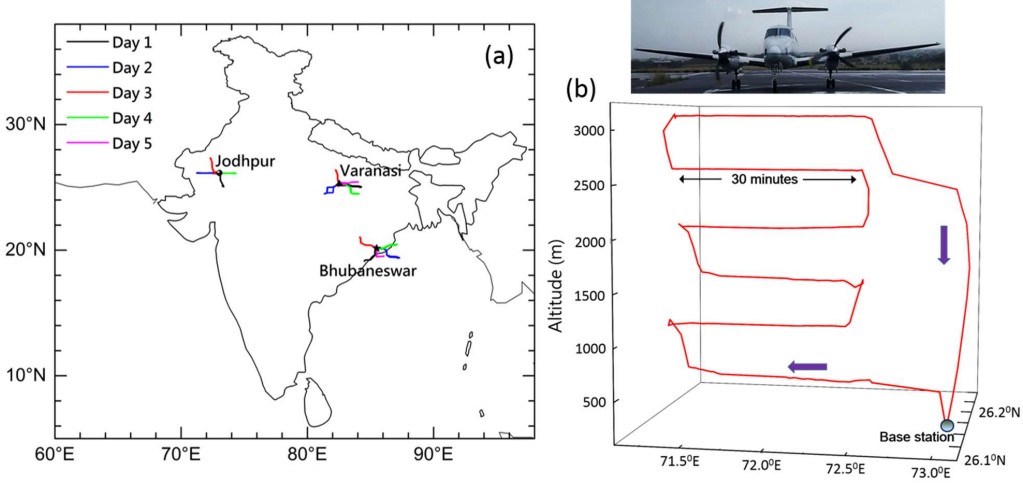

**Figure 2:** (a) Base stations for the aircraft sorties with the track of each sorties superimposed, (b)
A typical sortie pattern (staircase) which represents all the sorties carried out during the
experiment, and the photograph of NRSC aircraft.

### 2.2. Measurements

15   Ambient air was aspirated to the instruments using a solid diffuser inlet (University of Hawaii)

maintained at isokinetic flow conditions, as detailed in Babu et al., (2016), with a volumetric flow

rate of 70 LPM (litres per minute), for the average cruising speed of 300 km h$^{-1}$ of the aircraft. The

efficacy of the inlet to sample aerosols below 4 μm, under such conditions, has been demonstrated

during the DC-8 Inlet Characterization Experiment (*McNaughton et al., 2007*). Further details of

20   the experimental setup are explained in *Babu et al., (2016) and Vaishya et al., (2018)*. The air,



aspirated through this inlet, is then fed to different instruments through a manifold. Aerosol instruments onboard were calibrated prior to and after the campaign to ensure consistency in the measurements. Concurrent time and space coordinates were logged continuously using a high-resolution global positioning system (GPS).

CCN concentration at different supersaturations were measured at every second using a continuous

flow CCN counter (CCN-100 model by Droplet Measurement Technologies), by feeding the aspirated air continuously to the cylindrical column of the counter at a constant flow rate of 0.5 LPM, where it is exposed to desired supersaturations. Details of the principle of operation of the CCN counter are available elsewhere (*Roberts and Nenes, 2005; Lance et al., 2006*). Aerosols, according to their composition and size, having a critical supersaturation less than the effective

supersaturation inside the column, will spontaneously grow into a droplet as they exit the column. These droplets are counted with an optical counter using a laser of 650 nm wavelength. During each set of measurements, the supersaturation was varied through 0.2, 0.3, 0.4, 0.7, and 1.0% over a cycle of 30 minutes, and the cycle is repeated at each altitude level so that a complete CCN spectra (of CCN vs supersaturation) is available at every altitude level. In the present study, the

CCN concentrations never exceeded 5000 $cm^{-3}$ and hence the correction for water vapour depletion (*Lathem et al., 2011*) is not applied. Pressure correction was done to the set supersaturation at each altitude layer depending upon the change in pressure between ambient and calibration pressure (*Lance et al., 2009*). Data points during supersaturation transition are excluded due to the inherent ambiguity in the stability of the attained supersaturation. The measured CCN concentration has a

maximum uncertainty of 10 % (*Rose et al.,2008).*

Total aerosol number (CN) concentration was measured using an Ultrafine Condensation Particle Counter (Model 3776, TSI), developed by *Stolzenburg and McMurry, (1991)*. It measures CN of diameter 2.5 nm and above, with a time base of 1 minute. The aspirated air is continuously fed at 1.5 LPM, mixed with clean sheath air, which is saturated with butanol vapour while passing

through a saturator. The resultant flow is passed through a condenser where a sudden cooling result

in the condensation of butanol vapor onto aerosols due to supersaturation and the droplets are

counted using a counter working with a laser diode at 650 nm. Further details of the instrument

and its adaptability for aircraft-based experiments are explained by *Takegawa et al., (2017)*.

Aerosol absorption measurements at 7 different wavelengths (370, 470, 520, 590, 660, 880, and

950 nm) were carried out using a dual spot Aethalometer (AE 33 model of Magee Scientific)

(*Drinovec et al., 2015*) which works on the principle of filter-based optical attenuation technique

(*Hansen et al., 1984*). Filter loading artifact of the instrument is corrected in real time as explained

by *Drinovec et al., (2015)*. Absorption measurements were corrected for change in flow rate at

high altitudes following *Moorthy et al., (2004)*. Optical attenuation at 880 nm is used to estimate

the black carbon (BC) mass concentration using the specific absorption cross section value (7.77

$m^2$ $g^{-1}$). The Integrating Nephelometer (3563 model of TSI) measured the scattering coefficient

($\sigma_{sca}$) at 450, 550, and 700 nm wavelengths. Scattering measurements were corrected for non-

linearity in the angular truncation error following *Anderson et al., (1998)*.

For the CCN data analysis, initial five minutes of data at each altitude level were discarded

considering the stability of the measurements and the data was averaged for every minute. Hence

a minimum of 20 minutes of usable data comprising 5 supersaturations is available for each altitude

level. CN, spectral scattering, and spectral absorption measurements were also synchronized to the

1-minute averaged CCN data. Thus, for each region (East, west, and central IGP), 5 vertical

profiles of CCN and CN concentrations, and scattering and absorption coefficients were obtained.

**3.    Results and Discussions**

**3.1. Vertical distribution of CN and CCN**



Vertical profiles of CN and CCN concentrations (at ss= 0.4%) for the three sub-regions of the IGP are shown in Figure 3. Each profile is an average of all the sorties carried out from the base station. Significant differences are seen below ~ 1.5 km, which represents the well-mixed region within the PBL, and are attributed to the sub-regional scale emissions. As such, the CN concentrations are up by nearly a factor of 2 at the Central IGP (VNS) compared to the eastern or western ends

of the IGP; owing to the large-scale anthropogenic activities in the central IGP. Beyond ~ 2 km altitude, the CN concentrations remain quite comparable in magnitude, across the entire IGP with similar vertical variations.

In contrast to this, there is a significant difference in the aerosol type across the IGP (attributable to the source-heterogeneity), as revealed by the CCN concentration in the right panel of the same

Figure; especially in the free troposphere (above 2 km). Near to the surface, where the local source impacts dominate, the CCN concentration is the least over arid western IGP (JDR), followed by the industrialized Eastern IGP (BBR), with the VNS depicting the highest concentration. At all the sub-regions, CCN concentrations decrease towards higher altitudes. However, there is a sharp difference in the decreasing pattern; with the concentrations over the VNS falling off very rapidly

and almost merging with the profile over the arid region (JDR), the decrease is rather inconspicuous over BBR. The CCN concentration, though decreases initially with height up to 1 km, it was more or less steady above 1 km, suggesting prevalence of more hygroscopic particles aloft.

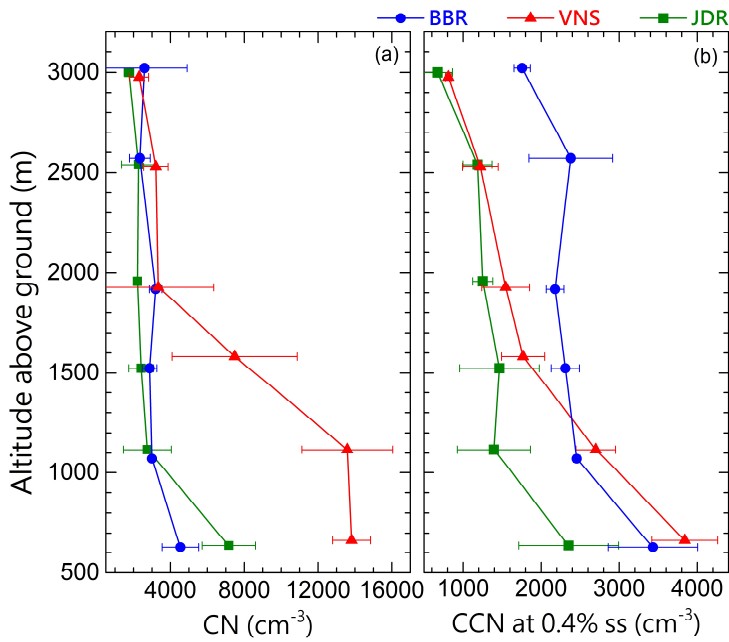

**Figure 3:** Vertical distribution of regionally averaged (a) aerosol number concentration (CN), (b) CCN number concentration, across the IGP. The symbols circle, triangle, and square represent BBR, VNS, and JDR respectively. Error bars represent the standard deviation around the mean values.

Irrespective of all these, the CCN concentrations remain high (1000 to >2000 cm⁻³ at 0.4% ss), even at 3 km altitude, which is above the base of monsoon clouds (*Das et al., 2017*). This will have strong implications in cloud modification, as has been established elsewhere (*Andreae et al., 2004; Rosenfeld et al., 2008*); however, their influence on the monsoon rainfall over the study region has not yet been quantified. Based on the aircraft observations during the CAIPEEX, over Hyderabad (17.45° N, 78.38° E) in southern India, *Padmakumari et al., (2017)* reported the suppression of warm rain process due to the presence of high CCN concentration. During the collaborative Regional Aerosol Warming Experiment (RAWEX) and the Ganges Valley Aerosol Experiment (GVAX), *Gogoi et al., (2015)* have reported CN and CCN (0.46% ss) concentrations



of ~ 2500 and ~ 1100 cm⁻³, respectively, for June 2011, from a high-altitude station (~ 2 km
a.m.s.l.), Nainital in Central Himalayas. The high CN and CCN concentrations observed in this
study is in line with values reported from Nainital, which is an optimal high-altitude site to study
regional (IGP) as well as transported aerosol characteristics over the IGP. In another study over
the Loess plateau in China during July-August months, *Li et al., (2015)* have reported high
concentrations of CN and CCN; peaking within the PBL and decreasing with increasing altitude.
*Lance et al., (2009)* have reported a CCN number concentration varying from ~ 200 to more than
10000 cm⁻³ during Gulf of Mexico Atmospheric Composition and Climate Study (GoMACCS)
aircraft campaign, over a heavily polluted region due to power plants and ship channels of
Houston. Local aerosol sources have a major role in determining the vertical structure during
period when high convective mixing prevail; while advection has a strong influence on the spatial
variation of altitudinal distribution above the PBL. From Figure 1, it is clear that there is an
advection of marine airmass near to the ground level (975 hPa) at both, east and west, regions of
the IGP, and intruding to the central IGP. However north westerlies from the continental region
pass through the free-tropospheric heights (700 hPa) of central IGP before reaching the east coast.
In short, the CCN concentration at cloud forming heights, which is a key parameter in deciding
the cloud droplet number concentration, is quite abundant over the IGP; decreasing spatially from
the eastern IGP to the western IGP especially above the PBL.

### 3.2. Altitudinal dependence of CCN – CN association

Aerosol number-size distribution and composition are known to show vertical variations (*Zhang
et al., 2011; Li et al., 2015*). Hence it is imperative to examine altitudinal dependency of CCN on
CN and its region-specific nature. In Figure 4, the altitude variation of the CCN-CN relationship
is presented for a constant supersaturation (0.4%), as a scatter plot of CCN vs CN, for the eastern
(top panel), central (middle panel) and western IGP (bottom panel) regions, respectively. Each
point in the Figure corresponds to the mean concentration at a particular altitude level above the



ground (identified by the colour) for each day of observation (identified by the shape of the point). The striking linear relationship over the entire altitude range at eastern IGP (BBR; top panel) clearly indicates the vertical homogeneity in aerosol composition in this region. The deviation of couple of points at highest altitude from this relationship indicates the presence of different aerosol types aloft. This is supported by the air mass back-trajectories which is examined in Section 3.4.

The CCN-CN relationship is quite nebulous over the Central IGP, which is a hotspot of anthropogenic activities, as revealed by the large scatter of the points in the middle panel. The scatter at the lower altitudes indicates influence of local source impacts, which also leads to large variation in the concentration as revealed by the large standard deviations. The association becomes better and stronger again as we move to the western IGP (JDR) where mineral dust is the

most dominant constituent.

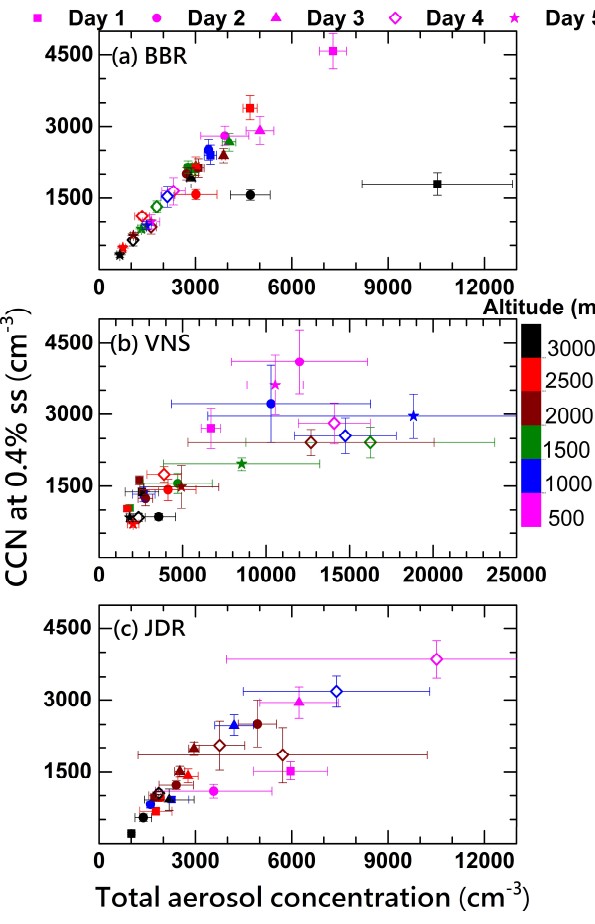

**Figure 4:** Association between the total CN and CCN number concentration at 0.4% supersaturation for (a) east IGP (BBR), (b) central IGP (VNS) and (c) west IGP (JDR) for each day of observation at all the observation heights. Colour code indicates the altitude above ground level while symbols represent the day of observation. Error bars represent the standard deviation around the mean values.

To further investigate the above hypothesis of the role of local emissions in weakening the relationship between CCN and CN over Central IGP, the variation of CCN number concentration at 0.4% supersaturation with BC mass concentration is examined in Figure 5. For this, the concurrent BC mass concentration measurements carried out from the same platform is used.



Central IGP showed highest absorption coefficient (column averaged) of $26 \pm 9$ Mm$^{-1}$, followed

by west ($16 \pm 2$ Mm$^{-1}$) and east ($15 \pm 3$ Mm$^{-1}$) IGP (Vaishya et al., 2018). It is interesting to note

that  the linear relationship is maintained for low to moderate concentrations of BC (up to around

1000 ng m$^{-3}$, which occurs mostly above PBL), while significant scatter occurs for higher values

of BC (exceeding 2000 ng m$^{-3}$), which occurs mostly in the lower altitudes, supporting the

hypothesis.  Similar deviations in CCN – CN relationship with respect to altitude has also been

reported by *Srivastava et al., (2013)* over the central IGP region, using aircraft measurements,

where they attributed it to the impact of local anthropogenic emissions.

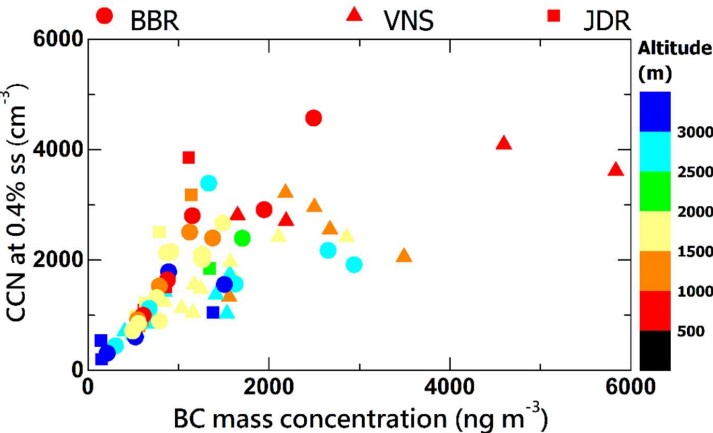

Figure 5: Association of CCN number concentration at 0.4% supersaturation with BC mass

concentration over the east - BBR (circle), central - VNS (triangle), and west - JDR (square) IGP

regions. Colour code indicates the altitude of observation.

**3.3. CCN spectra and parameterisation for different altitudes**

Using the measurements of CCN number concentration as a function of supersaturation the mean

CCN spectra are constructed, for different sub-regions of IGP, and is shown in Figure 6 for

different altitudes. In addition to the regional distinctiveness in the CCN number concentrations

seen in Figure 3, it is interesting to note the rapidly levelling off of the spectra with increasing





5     supersaturation, at the eastern IGP (represented by BBR, blue lines in Figure 4), especially above

1 km; in contrast to the other two regions, where the CCN concentrations keep on increasing with

increasing supersaturation at all heights. This clearly demonstrates a change in the hygroscopicity

of aerosols across the IGP, especially in the free-troposphere. To quantify this, the CCN spectra

are parameterized by evolving a least square fit with Twomey's relation (*Twomey 1959*),

$$CCN(ss) = C(ss)^k \qquad (1)$$

where, CCN (ss) is the number concentration of CCN at a particular super saturation (ss), C and

k are empirical coefficients. The shape of the CCN spectra, represented by the 'k' values, showed

significant altitudinal variations. The altitude variations in the CCN spectra, which can be due to

the variations in aerosol number size distribution, will have an impact on the droplet size

15    distribution of the warm cloud formation (*Raga and Jonas., 1995*).

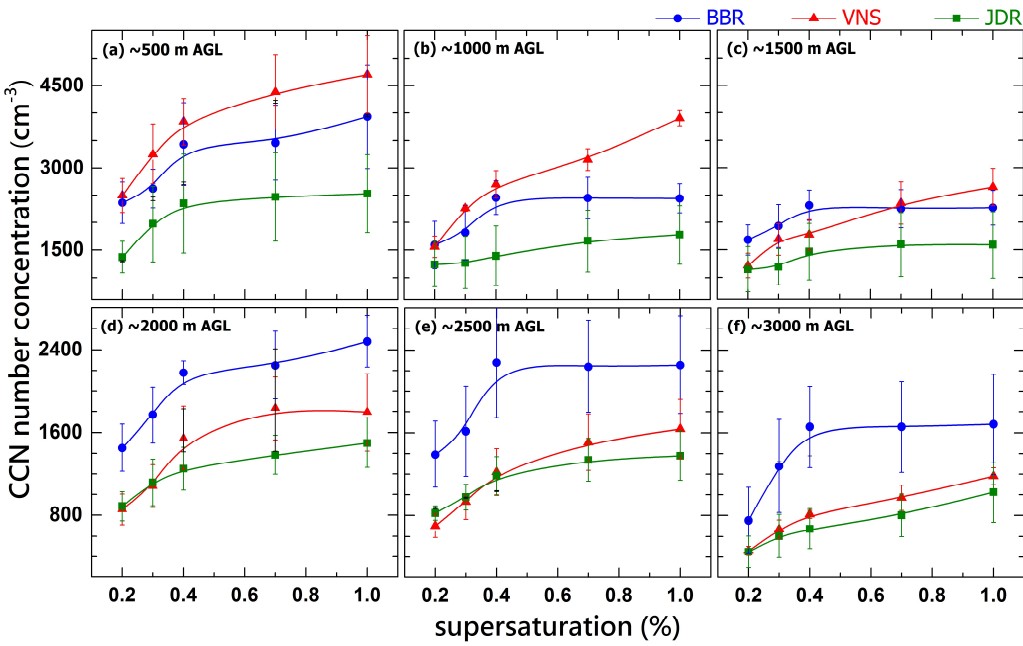



**Figure 6:** Mean CCN spectra at six altitudes levels over different sub-regions of the IGP. The error bars indicate the standard deviation around the mean. The points correspond to measurements, while the lines are the empirical fits; circle, triangle and square representing respectively the east (BBR), central (VNS), and west (JDR) IGP sub-regions.

The vertical variation of the k values for each region is shown in Figure 7, which reveals a distinct transformation of the CCN properties of aerosol across the IGP. Over the eastern IGP (which is industrialized and has coastal proximity), k is the least; with a small vertical variation that shows a weak decrease initially and then a weak increase. The arid western IGP shows a very similar vertical variation of k; but the values remain consistently higher than those seen for the eastern IGP, at all heights. The highest values of k are seen over central IGP, with a steady increase with altitude. Across the entire IGP, k increases with altitude, indicating a decrease in the hygroscopicity with altitude or a rapid change in the number size distribution. As the CCN concentration at higher supersaturations (>0.4%) are mainly governed by the concentration of small particles (<~ 70 nm) *(Lance et al., 2009)*, the corresponding high CCN concentration suggests the presence of a prominent fine mode aerosol system, which is clearly seen over the entire IGP; especially over the central IGP. The near-flat CCN spectra at BBR (above 0.4% supersaturation) indicate the presence of highly soluble or coarse mode aerosols, such that almost all aerosols are activated at 0.4% supersaturation itself. Similar observations of low k values (~ 0.2) are reported by *Jayachandran et al., (2017)* from a coastal location in peninsular India during sea breeze regime of the monsoon season, when both the local mesoscale and synoptic circulations bring marine (seasalt) aerosols to the region.



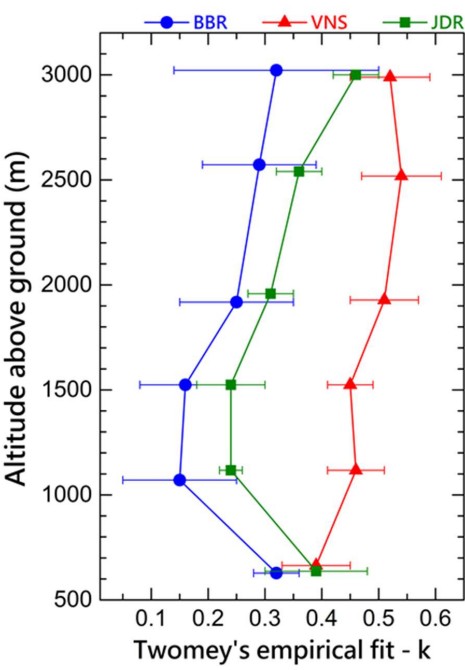

**Figure 7**: Altitude variation of k (Twomey's empirical fit) over east - BBR (circle), central - VNS (triangle), and west - JDR (square), IGP regions. Error bars represent standard deviation of the fit.

The similarity in the vertical profiles of k-value over the west and east regions of the IGP show presence of similar nature of CCN active aerosols over both the regions. The reported CCN spectra and k values over the Indian sub-continent at higher altitudes are listed in Table 2, for different aerosol types using both fixed and aircraft-based platforms. It should be noted that the k values depend on the supersaturation range used for its estimation and hence the supersaturation range is also mentioned in the table. From the Table, it can be seen that the values reported from central Himalayas (2 km a.m.s.l.) are similar to the present observations over central IGP at similar altitudes. Central Himalayas experience airmass from IGP as well as semi-arid regions of west Asia during pre-monsoon, and *Dumka et al., (2015)* have reported mean k value of ~ 0.58 for June 2011 during the RAWEX-GVAX campaign. The current observations show k values above 0.51 for altitudes above 2 km over Central IGP. In the present study, the k values estimated for the



5 altitude 2 - 3 km a.g.l., are in the range 0.25 - 0.32 and 0.31- 0.46 above east and west IGP

respectively. *Roy et al., (2017)* reported a mean k value of ~ 0.38 during pre-monsoon over eastern

Himalayas, when airmass reached the site from IGP as well as semi-arid regions of west Asia.

Examining the CCN spectra at cloud base (~ 1600 m) during the CAIPEEX campaign (October,

2011) over peninsular India, *Varghese et al., (2016)* have reported high k values (0.72) associated

10 with polluted conditions and low k values (0.25) during clean conditions. Flat CCN spectra having

low 'k' values observed in this study over east and west IGP indicate high CCN active nature of

the aerosols.

**Table 2**: Reported k values and supersaturation (ss) range used for the estimation, along with the

CCN (at 0.4% ss) concentrations, for high altitudes above the Indian sub-continent.

| Sl. No. | Location (Lat, Long) | Type (mode) | Altitude, (a.m.s.l., km) | Period | $CCN_{0.4}$ (cm$^{-3}$) | k (ss range) | Reference |
|---|---|---|---|---|---|---|---|
| 1 | Eastern IGP | Polluted marine | | | ~ 2200 | 0.25 (0.2-1.0) | |
| 2 | Central IGP | polluted | 2 | June 2016 | ~ 1500 | 0.51 (0.2-1.0) | Present study |
| 3 | Western IGP | Semi-arid | | | ~ 1250 | 0.31 (0.2-1.0) | |
| 4 | Eastern Himalayas 27°N, 88.2°E | Urban (fixed) | 2.2 | Mar-May, 2016 | ~1800 | 0.38±0.1 (0.1 – 1.0) | Roy et al., 2017 |



| | | | | | | |
|---|---|---|---|---|---|---|
| 5 | Central Himalayas 29.4° N, 79.5° E | Background (fixed) | 2 | June, 2011 | ~1000 | 0.57±0.11 (0.17 – 0.75) | Dumka et al., 2015 |
| 6 | Eastern IGP | Polluted marine | | | ~ 2300 | 0.16 (0.2-1.0) | |
| 7 | Central IGP | polluted | 1.5 | June 2016 | ~ 1800 | 0.45 (0.2-1.0) | Present study |
| 8 | Western IGP | Semi-arid | | | ~ 1500 | 0.24 (0.2-1.0) | |
| 9 | Hyderabad | Polluted (aircraft) | 1.5 | October 2011 | ~1100 | 0.72 (0.2 – 0.8) | Varghese et al., 2016 |
| 10 | | Clean (aircraft) | | | ~500 | 0.25 (0.2 – 0.8) | |
| 11 | Mahabaleshwar 17.6°N, 73.4°E | Western Ghats | 1.4 | Mar – May, 2013 | ~1500 | 0.5 (0.2 – 1.0) | Leena et al., al., 2016 |

### 3.4. CCN Activation Efficiency: Vertical structure and variation across the IGP

CCN activation efficiency is the ratio of CCN number concentration at a particular supersaturation to the total CN concentration. This ratio has been estimated as a function of altitude for each of the sorties and the mean vertical profiles are shown in Figure 8a for 0.4% supersaturation. Similar

10 to the altitude variation of k shown in Figure 7 over distinct regions of IGP, the activation efficiency is the least over the Central IGP (VNS), and the most in the eastern IGP (BBR) with JDR coming in-between. At all the stations, the efficiency remains low within the PBL (below1.5



km) where the local source impacts are rather strong.  Above the PBL, it either increases or remains

steady with altitude before decreasing again above 2.5 km, probably due to different aerosol types

(less hygroscopic, finer particles) at the higher levels. The low CCN efficiency over VNS is

associated with the presence of higher concentration of BC (>4000 ng m$^{-3}$) and CN number

(>10000 cm$^{-3}$), indicating a pollution surrogate from anthropogenic sources modifying the CCN

activation.

The variation of k with CCN activation efficiency at 0.4 % supersaturation for east IGP (BBR,

circle), central IGP (VNS, triangle) and western IGP (JDR, square) are shown in Figure 8b. High

values of k are observed with low CCN activation efficiency and vice-versa, showing an inverse

relationship between the two parameters. CCN efficiency and k over the desert region vary from

~ 20 % to 65% and ~ 0.2 to 0.7, respectively.  Similar inverse association between CCN efficiency

and k is reported by *Hegg et al., (1991)* and *Jayachandran et al., (2017)*. High k values are due to

the dominant presence of small or less soluble particles in the aerosol system, which in turn reduce

the CCN efficiency.  In central IGP, these points (low CCN efficiency and low k values) which

deviate from the inverse relation are observed within the PBL. At high altitudes (>3 km) over the

IGP, *Srivastava et al., (2013)* have reported aerosol size distribution peaking below ~ 40 nm due

to new particle formation (NPF) events and cloud processing. *Rose et al., (2017)* have reported the

significant role of NPF in CCN activation above PBL especially during wet season at Chacaltaya

(5240 m a.m.s.l), Bolivia. In the present study, the role of cloud processing or in-cloud scavenging

for low CCN efficiency and flat CCN spectra (low k) at cloud forming heights cannot be neglected.





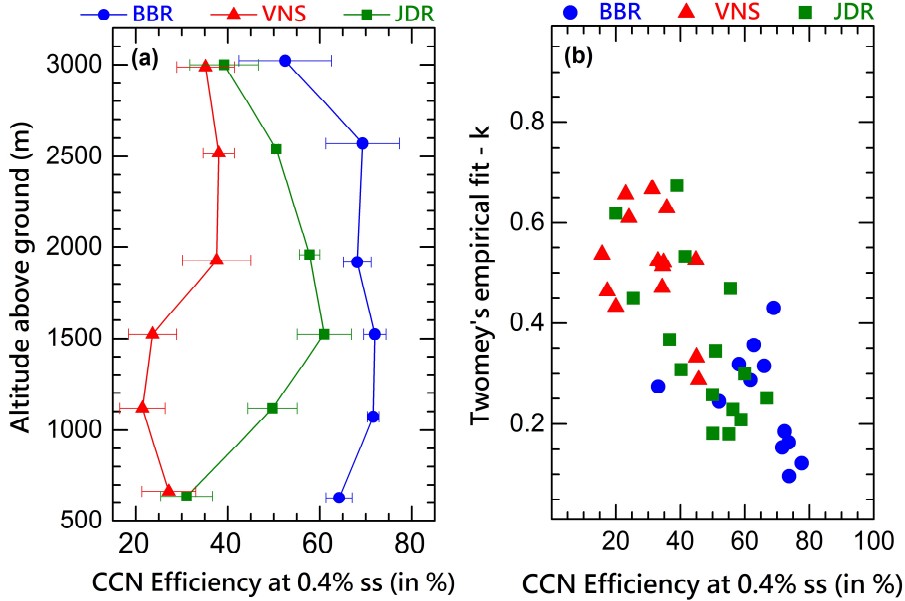

**Figure 8:** Vertical distribution of (a) mean CCN activation efficiency at 0.4% supersaturation, and (b) variation of k values with the corresponding CCN efficiency at 0.4% supersaturation over east - BBR (circle), central - VNS (triangle), and west - JDR (square) IGP regions. Error bars represent standard error of the mean.

10 Based on measurements at the mean sea level and at 1 km above ground level, *Jayachandran et al., (2018)* have shown the vertical heterogeneity existing in CCN efficiency and CCN spectra during the ISM at the south coast of India. *Li et al., (2015)* have shown that the anthropogenic influences can cause a strong variation in CCN efficiency from 10% to 70% from near ground level to about 4.5 km over China during Asian summer monsoon season. More than 50% of the

15 aerosols are CCN active over the regions other than central IGP, which indicates the dominant role of natural aerosols in warm cloud droplet activation over the sub-continent region just prior to the ISM season. The airmass traversing through the polluted-continental regions is responsible for the lowering of CCN activation efficiency at the free troposphere heights over east IGP. Significant influence of the nature of airmass on CCN activation over the Indian region is illustrated by the


closure studies carried out by *Srivastava et al., (2013)* at various altitudes. *Jayachandran et al., (2017)* have reported higher CCN activation efficiency for marine airmass than continental from ground-based observations from peninsular India during the ISM. Within the PBL including near to the ground level, CCN efficiency is very high over the east IGP (coast) which will support the cloud droplet formation with a sharp droplet size distribution.

One of the striking features emerging from this study is the high CCN efficiency over the arid region of Western IGP, which is reported for the first time. This region is known for its dust dominance (both locally generated and advected from the Middle East and Eastern Africa). Though pure dust is water inactive, its CCN efficiency will enhance when coated or mixed with soluble salts like sulphates, nitrates etc. *(Zhang et al., 2006; Kelly et al., 2007)*. Though *Feingold*

*et al., (1999)* have shown that coarse mode dust aerosols can act as giant CCN and initiate drizzle formation, their number concentration is far less numerous, especially at high altitudes (*Padmakumari et al., 2013*). Thus, the observations of moderately high CCN activation efficiency, lower values of k and higher concentration of CCN are interesting and need discussions. Figure 9 shows airmass back trajectories for five days and arriving at 500 m, 1500 m, and 3000 m a.m.s.l

above (a) east IGP - BBR, (b) central IGP - VNS, and (c) west IGP - JDR. From Panel (c), it can be seen that the airmass reaching JDR (conducive for dust-advection) has significant history over the northwestern Arabian Sea, and hence would also carry significant moisture. It is known that presence of hygroscopic salt aerosols can catalyse the reaction of dust with acidic gases *(Tobo et al,, 2010)*, changing its hygroscopicity. Thus, the airmass reaching the desert region, having a

strong marine component could enhance the activation efficiency of the aerosols. Strong convection at the lower atmosphere will also take salt aerosols to the atmosphere from the regional dry salt lakes. *Begue et al., (2015)* have reported CCN efficiency of ~70% for 0.2% supersaturation over Netherlands during a dust transport event due to the accumulation of solute particles on dust.

Present study shows that about 66% of the total aerosols in the mixed layer of western IGP - JDR

were activated as CCN at 1% supersaturation.

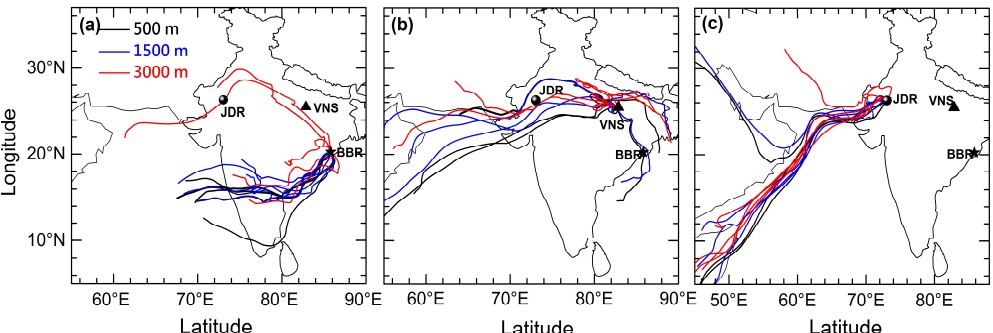

**Figure 9:** Five days airmass back trajectories at 500 m (black), 1500 m (blue) and 3000 m (red)
a.m.s.l over (a) East IGP (BBR), (b) Central IGP (VNS), and (c) West IGP (JDR) during the
campaign period.

The coastal location BBR (panel a,) is strongly under the influence of marine airmass, though has

considerable travel across the Indian mainland initially, enters the Bay of Bengal, turns and then

arrives at BBR, thus would be moisture laden and would contain sea-salt particles. On the other

hand, at the Central IGP, irrespective of the history of the airmasses, they have to travel

considerable distance across the mainland, and are thus conducive for advection of anthropogenic

aerosols, besides losing a significant amount of moisture it had acquire from the ocean. Thus, VNS

is under the influence of local emissions, which include hydrophobic particles such as BC, which

is also in the fine size range, while the air is deprived of moisture; all of them resulting in the

highest values of k and lowest values of CCN activation efficiency of the three IGP sub-regions.

Vertical profiles of CCN efficiency over VNS for the first day of observation (8 June) when the

airmass was from the marine region (Bay of Bengal), and the mean picture for the other days

(when the airmass was continental) are shown in Figure 10a (left panel), respectively by dotted





and continuous bold lines. The significant increase in the activation efficiency during marine

airmass conditions is very conspicuous.

At BBR, there have been two episodes of pre-monsoon precipitation on 4, 5 June 2018 (much

before the sortie timings), with accumulated rainfall of 58 and 8 mm; and at VNS a rainfall of 20

mm occurred on the evening of 7th June. The vertical profile of CCN activation efficiency over

BBR averaged for measurements before and after rainfall is shown in bold and dotted lines,

respectively, in Figure 10(b). There is a decrease (though weak) in the activation efficiency

(especially below the cloud level, 2 km), after the precipitation, probably due to removal of

hygroscopic aerosols by the precipitation. Near the ground level, CCN concentration (mean ±

standard deviation) reduced from $3431 \pm 572$ to $1320 \pm 454$ cm$^{-3}$ and from $1755 \pm 105$ to $460 \pm$

$209$ cm$^{-3}$ at ~3 km a.g.l. After the rainfall, a reduction (<10%) is seen in the CCN efficiency over

BBR, meanwhile, there is a large diminution in the number concentration of CN and CCN.

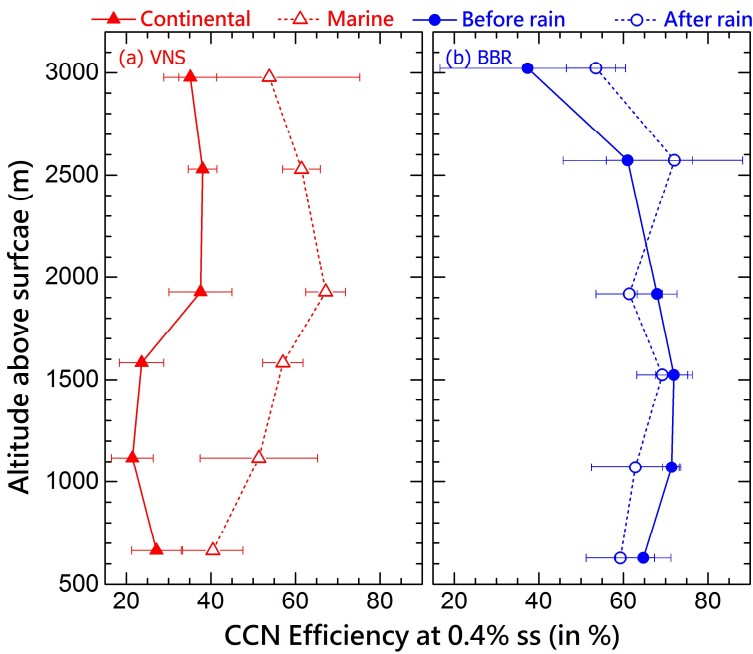



**Figure 10:** Altitude variation of CCN efficiency at 0.4% supersaturation showing the effect of (a) distinct airmass, continental – solid line and marine – broken line, at central IGP – VNS, and (b) rainfall, before – solid line and after – broken line, at east IGP - BBR.

The theoretical framework of wet scavenging process accounts for nucleation, gravitational and inertial impactions, and turbulence scavenging mechanisms *(Pruppacher and Klett, 1997)*. But uncertainties and difficulties still exist in attributing the observational evidences of wet scavenging of aerosols to different scavenging mechanisms, especially in the case of moving air parcels. Efficiency of below-cloud scavenging (wash out) mainly depends on the number size distribution of both aerosols and raindrops, while the in-cloud scavenging (rain out) depends mainly on the solubility of the aerosols *(Garrette et al., 2006)*. Studies *(Andronache, 2003)*, suggest that in-cloud scavenging is more effective in the removal of CCN, rather than below-cloud scavenging. The decrease in CCN concentration over BBR after the rainfall, and the high CCN efficiency seen in the present study indicates the highly soluble nature of aerosol system prevailing over the region. The modification in CCN efficiency over VNS and BBR underlines the role of type of airmass and rainfall in determining the vertical structure of CCN activation in a short duration.

## 3.6. CCN and aerosol optical properties

Concurrent measurement of aerosol scattering and absorption coefficients during the campaign provided an opportunity to examine possible links between CCN and the optical properties of aerosols. *Liu and Li, (2014)*, and *Jefferson (2010)* have illustrated the potential of using aerosol optical properties as a proxy and prognostic variable for studying the CCN properties. *Liu and Li, (2014)* have used the scattering aerosol index (AI), which is the product of scattering coefficient (at 450 nm) and scattering Angstrom exponent to link the aerosol scattering properties to CCN concentration. Following their approach, we have estimated AI as $AI = \sigma_{sca}(450) \times \alpha_{sca}$ where $\sigma_{sca}(450)$ is the scattering coefficient at 450 nm, estimated from the Nephelometer data, and $\alpha_{sca}$





is the Angstrom exponent, estimated over the wavelength range 450, 550 and 700 nm by evolving

a least-squares fit to the relation

$$\sigma_{sca}(\lambda) = \sigma_0 \lambda^{-\alpha_{sca}} \qquad (2)$$

The scatter plots of CCN concentration at 0.4% supersaturation against scattering AI are shown in

Figure 11, with panels from left to right representing Eastern, Central and Western IGP, along with

the corresponding altitudes of measurement, indicated by the colour code. Linear least-squares fits

to the points through the origin (implying that all the scattering aerosols contribute to CCN

concentration), are also shown in the Figure along with the fit parameters. Very good linear

dependencies emerge from the Figure, for all the stations across the IGP, though the slope appears

to be region specific.

The highest slope is observed at the least anthropogenically impacted / dust dominated Western

IGP, while the least value is shown at the Central IGP, which has the highest concentration of

absorbing aerosols. As AI is a product of scattering coefficient and scattering Angstrom exponent,

it carried signatures of total particulate loading, and the size distribution.

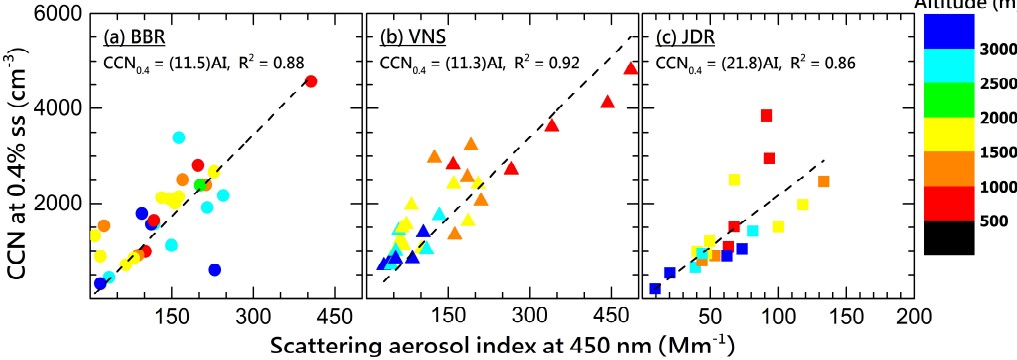

**Figure 11:** Association between the total scattering AI at 450 nm and CCN number concentration

at 0.4% supersaturation for (a) eastern (BBR) (b) central (VNS), and (c) western (JDR) IGP





regions. Color indicates the altitude of measurement. Dashed lines represent linear least-squares

fit to the points for each region. Regression slopes and squared correlation coefficients are written

in each panel. Note that the X-axis scale is different for the right-most panel.

Examining this Figure, along with the CN concentration shown in the left panel of Figure 3, it

appears that the higher slope at JDR is most likely due to the large size dust aerosols there, because

the coarse size distribution would lead to a smaller Angstrom exponent, because CN concentration

at JDR is very well comparable to that at BBR, except at the lowest altitude of 500 m, implying

that the total scattering coefficients would have to be of comparable magnitude. But the slope at

BBR is nearly half of that seen at JDR, despite it having the highest activation efficiency. On the

similar lines, it appears that the size distribution of aerosols over VNS has more fine particles

(higher Angstrom exponent, but lesser activation efficiency). Thus, the size distribution and

chemistry of the aerosol influence the relation between scattering aerosol index and CCN

concentration. This dependency is useful in developing empirical relationship connecting CCN

and light scattering properties at least in a region-specific scale. The number concentration of

Aitken mode aerosols, especially the aerosols at 60-100 nm range and its composition is the main

factor in governing the variability in CCN properties, while the relative dominance of

accumulation mode aerosols will be determining the scattering properties. Figure 11 demonstrates

the strong relationship that exists between the aerosol scattering properties and CCN concentration

in the vertical column over the IGP. This would facilitate estimating the CCN profiles from the

vertical profiles of aerosol scattering properties.

**4.  Conclusions**

Extensive characterisation of the altitude distribution of CCN and its spatial variation across the

IGP has been carried out, for the first time, using in-situ measurements aboard an instrumented

aircraft just prior to the onset of the Indian summer monsoon (ISM). The results concluded below



5     form a significant step towards the ACI during the Indian Summer Monsoon, though the impact on cloud microphysics needs further investigation.

- Spatial heterogeneity in total aerosol concentration exist over the IGP with high concentrations (>13000 cm$^{-3}$) over the central IGP (near to the ground level) and the least over the western IGP while it's vertical variation remain the same above the planetary boundary layer at all regions.

- High CCN concentration (above 1000 cm$^{-3}$ at 0.4% supersaturation) is observed up to 2.5 km across the IGP, indicating significant possibility of aerosol indirect effects.

- Central IGP shows higher CCN activation efficiency above the planetary boundary layer (>1.5 km), than within, despite the latter having high CN and CCN concentrations indicating activation of aerosols as CCN is suppressed by freshly emitted aerosols, mostly from anthropogenic sources.

- High CCN activation efficiency, ~61% at 0.4% supersaturation, at ~1.5 km above the ground level is observed over the dust dominated western IGP.

- It is seen that while precipitation reduces the CCN activation efficiency below cloud level, advection of marine airmass enhances CCN efficiency, even over arid regions.

- An empirical relationship between the CCN activation and scattering properties of aerosols has emerged as an important step towards CCN prediction over the region.

### Data availability

Data are available upon request from the contact author, S. Suresh Babu (s_sureshbabu@vssc.gov.in).

### Competing interests

The authors declare that they have no conflict of interest.

### Author contributions





SSB, SKS and KKM conceptualized the experiment and finalized the methodology. SSB, VJ, AV and MMG were responsible for the data collection onboard aircraft. VJ carried out the scientific analysis of the data supported by SSB, VSN and AV. VJ drafted the manuscript. SSB, KKM and SKS carried out the review and editing of the manuscript.

**Acknowledgments**

This study was carried out as part of the SWAAMI-RAWEX (South West Asian Aerosol Monsoon Interaction – Regional Aerosol Warming Experiment) campaign. We thank Director, National Remote Sensing Centre (NRSC), Hyderabad and the Aerial Services and Digital Mapping Area (AS & DMA) for providing the aircraft support for this experiment. Aditya Vaishya was supported

by the Department of Science and Technology, Government of India, through its INSPIRE Faculty Programme. Details of the aircraft data used in the present study and the point of contact are available at http://spl.gov.in; "Research Themes;" "Aerosol, Trace gases and Radiative Forcing Branch." The RAWEX project is supported by ISRO (Indian Space Research Organisation) and the SWAAMI project is supported by MoES (Ministry of Earth Science).

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
