# Peer review of "Altitude profiles of CCN characteristics across the Indo-Gangetic Plain prior"

_Atmospheric Chemistry and Physics, 2019_

## Referee Comment (RC1) · Anonymous Referee #1 · 26 Aug 2019

General comments:

The submitted manuscript reports and discusses the detailed airborne measurements of cloud condensation nuclei (CCN) and aerosol scattering/absorbing properties conducted during 2016 pre-monsoon over the Indo-Gangetic Plain (IGP) region covering urban-industrial, semi-arid, and coaster areas. The analysis of the airborne datasets finds the highest CCN concentration over the central IGP accompanied by least CCN activation efficiency possibly linked to higher amounts of with high black carbon (BC). Similar measurements over western semi-arid IGP show high CCN efficiency indicating the hygroscopic nature of mineral dust particles. The vertical structure of CCN

reflects the role of marine airmass in increasing CCN efficiency. Finally, the study reveals an empirical relationship between CCN and aerosol scattering properties that can potentially predict the CCN from aerosol optical properties.

The paper brings out important detailed information on CCN and aerosol properties over the monsoon region of northern India where the complex aerosol properties might have substantial impacts on clouds, thereby precipitation, through aerosol-cloud interactions. Overall, the paper is well-written, however, requires proofreading from an expert to improve the language and presentation further. The topic addressed in the article fits into the scope of ACP. The conclusions drawn based on dataset, methodology, and research analysis are reasonable and mostly clear to the reader.

Attached are several comments I have derived while reviewing the manuscript, which authors need to clarify during the revision process. The paper can be published after these corrections, and also those from other reviewers, are reflected in the revised manuscript.

Abstract Line 6-10: Too long statement. Break it into two. Line 15: "...followed by that in the west" Line 17: "...at all altitudes" Line 20: "The Great Indian Desert", also known as "The Thar Desert" Line 23-24: Due to washout of aerosols?

Page 4, line 5-10: Authors may cite following references related to the seasonality of aerosols (Jethva et al., 2005) and trends (Dey and Girolamo, 2011) over IGP.

Jethva, H., Satheesh, S. K., and Srinivasan, J. (2005), Seasonal variability of aerosols over the Indo-Gangetic basin, J. Geophys. Res., 110, D21204, doi:10.1029/2005JD005938.

Dey, S., and Di Girolamo, L. (2011), A decade of change in aerosol properties over the Indian subcontinent, Geophys. Res. Lett., 38, L14811, doi:10.1029/2011GL048153.

Page 4, line 16: "...one of the best natural laboratories for investigating the complex nature of aerosols on clouds and precipitation."

Page 5, line 5-7: The paragraph seems to be ending abruptly. Here, the author should mention concisely about the overall goals of the airborne experiments and objectives of the paper. Also, a brief writeup about how the analysis was conducted and what they were looking for would help the reader to familiarize with the overall content of the paper.

Page 5, line 12: "before the onset of ISM over central and northern India"

Page 5, line 13-14: "also shown in Figure 2"

Page 5, line 20: "...at the southern peninsular coast of Kerala state"

Page 7, line 5: "All aircraft sorties..."

Page 7, line 12-13: "Due to the unpressurized..."

Page 7, line 5-10: Wouldn't be good if another sub-plot of the ratio of CCN-to-CN is added here?

Page 13, line 13: "there seem to be a notable difference in the hygroscopicity of aerosols...". Author needs to be specific here though the hygroscopicity of aerosols is associated with aerosol type.

Page 15, Figure 4: The CN-CCN relationship over all three stations looks near-linear for the CCN range up to 3000-5000 cm-3, after which it becomes non-linear irrespective of the difference in aerosol type. Interesting.

Page 16, Figure 5: If possible, please reverse the colors in the scale, i.e., blue for lowest altitude, red for the highest.

Page 17, line 5: ...represented by BBR, blue lines in Figure 6 (?). Also, is the flattening of CCN curve with SS for BBR an indication of aerosol type/size. Due to its proximity to the coast, BBR is likely influenced by coarse sea-salt particles against finer size aerosols over interior IGP.

[Figure]

Throughout the discussion in this section, the author should cite corresponding aerosol studies supporting the link between aerosol type/size and CCN spectra.

Table 2. A concurrent geographical plot showing k values with colored circles as a symbol would be more effective and easier for a reader to grasp the reported values.

Page 28, line 15: slopes for BBR and VNS are comparable.

Page 28, Figure 11: Maintain the same X-axis range for all three sub-plots.

Page 28, line 17-18: While this assumption generally holds, it would be interesting to plot CCN as a function of extinction aerosol index. Since the aircraft measurements delivered both scattering and absorption coefficients, it would be straightforward to create a similar plot using extinction AI.

Page 29, line 23-24: The relationship between CCN and aerosol properties further implied the use of satellite-retrieved AOD products in the region, which are now matured and fairly accurate, and model-generated aerosol profile aided by ground (MPLNET at Kanpur)/space lidar (CALIOP), in predicting CCN. This should be mentioned here and also in the conclusion.

Also, did author check the relationship between CCN efficiency and aerosol scattering/extinction properties, if any? It is worth to perform such analysis.

Page 30, line 5: "...towards characterizing/understanding the ACI..."

─────────────────────

---

## Referee Comment (RC2) · Anonymous Referee #2 · 2 Oct 2019

General comments: The manuscript presents the altitude variations of CN and CCN characteristics in the troposphere below 3 km based on systematic aircraft based observations carried out over the western, central and eastern IGP during June 2016 , just before the onset of Indian summer monsoon. Altitude variations of CCN activation efficiency, CCN spectra and their similarities and differences over the three regions are investigated, in the light of the potentially different aerosol sources. One of the most striking features observed is the high CCN activation efficiency over the dust-dominated western IGP. The topic of research is highly relevant and current and the results presented here are of very high importance for a wide range of scientific community (including aerosols, aerosol-cloud interaction and climate modelling studies).

[Figure]

The manuscript is well written and the conclusions are well supported by observations.

I have a few minor comments/suggestions; addition/modification of a few sentences will be sufficient to address all these comments. Considering the very high scientific importance of the results presented, which are of interest to a wide scientific community, I recommend that the manuscript may be accepted for publication in ACP after minor revision.

Page-22, Paragraph-2 & Fig.8(b): This information is redundant as it can be inferred from Figs. 6, 7, and 8(a) and the discussions on them. However, if the authors want to still keep it, please spell out the following: Lines 18-19: what is the range of 'low CCN efficiency' and 'low k values' referred here? I think this result is not very evident in Fig.8(b).

Page-23, Lines 17-18: "The airmass traversing through the polluted-continental region is responsible for the lowering of CCN activation efficiency at the free troposphere heights over the east IGP". Note that, among the three regions considered here, the CCN efficiency is highest at all altitudes over BBR (e.g., Fig.8a). The above statement can be true if the CCN activation efficiency is found to be higher when the tropospheric airmass transport over BBR is from the east compared to those from the west. You may clarify how this conclusion was arrived at? Please delete the sentence if it cannot be explained unambiguously .

Page-24, second paragraph: This is a very interesting and, perhaps, the most important finding from this study. It has major implications in ACI.

Page-25, Line 18: "... while the air is deprived of moisture;...". Note that 'k' is related to the property of aerosols (size distribution and water affinity, as stated in the manuscript) and is measured by systematically changing the supersaturation inside the instrument. Then, how "depriving of moisture" in the atmosphere will result in high value of 'k'?

Page-26, Lines 8-9: "... and at VNS a rainfall of 20 mm occurred on the evening of

7th June". What is relevance of this statement here? Dependence of any aerosol/CCN parameter on rainfall at VNS is not presented in this manuscript.

Page-26, Line-15: "After the rainfall, a reduction (<10%) is seen in the CCN efficiency over BBR, ...". This is true for the height range below 2000 m, while the opposite is the case for 2500-3000 m (Fig.10). Is it because of the difference between in-cloud and below-cloud processes that remove/shift the size distribution? Also see the comment below. If it cannot be explained based on the present set of observations, please include a line on the differences (CCN efficiency) observed in the altitude range of 2500-3000 m.

Page-27, Lines 14-17: How this process (more efficient removal of CCN by in-cloud scavenging) can enhance the CCN efficiency after rainfall (height range of 2500-3000 m; Fig.10)?

Page-28: Lines 8-12: Modify this sentence (it is not very clear; contains 'because' twice).

Page-28, Lines 11-12: "... implying that the total scattering coefficient would have to be of comparable magnitude". Why this guess? You already have the scattering coefficient measurements available (used for estimatiing AI). Did I miss something?

Page-30 (Conclusions), Lines 17-18: "High CCN activation efficiency ... dust dominated western IGP". This is a very interesting and important result. A statement on its implication will be highly useful.

Other suggestions:

Page-4 Line 20: Keep proper reference format Page-5 Line 14: Change : "Synoptic wind ..." as "Monthly mean synoptic wind ..." Page-11, Line 1: expand "ss" (first time usage of ss for supersaturation) Figure 4: If you have sufficient ancillary data required, it would be interesting to know why there are major deviations from the general trend on (i) Day-1 at BBR and (ii) Day-4 at VNS. Is this the effect of rain or change in airmass

trajectory?

---

## Author Comment (AC1) · 10 Nov 2019

At the outset we appreciate the summary evaluation of the importance of our results by both the referees, and the recommendations. We have carefully considered their comments and suggestions and revised the paper accordingly. Our point-by-point responses to the comments, based on which the revisions are made, are given below. The review comments are given italics, while the author responses are in bold font

**Anonymous Referee #1**

**General comments:**

*The submitted manuscript reports and discusses the detailed airborne measurements of cloud condensation nuclei (CCN) and aerosol scattering/absorbing properties conducted during 2016 pre-monsoon over the Indo-Gangetic Plain (IGP) region covering urban-industrial, semi-arid, and coaster areas. The analysis of the airborne datasets finds the highest CCN concentration over the central IGP accompanied by least CCN activation efficiency possibly linked to higher amounts of with high black carbon (BC). Similar measurements over western semi-arid IGP show high CCN efficiency indicating the hygroscopic nature of mineral dust particles. The vertical structure of CCN reflects the role of marine airmass in increasing CCN efficiency. Finally, the study reveals an empirical relationship between CCN and aerosol scattering properties that can potentially predict the CCN from aerosol optical properties.*

*The paper brings out important detailed information on CCN and aerosol properties over the monsoon region of northern India where the complex aerosol properties might have substantial impacts on clouds, thereby precipitation, through aerosol-cloud interactions. Overall, the paper is well-written, however, requires proofreading from an expert to improve the language and presentation further. The topic addressed in the article fits into the scope of ACP. The conclusions drawn based on dataset, methodology, and research analysis are reasonable and mostly clear to the reader.*

*Attached are several comments I have derived while reviewing the manuscript, which authors need to clarify during the revision process. The paper can be published after these corrections, and also those from other reviewers, are reflected in the revised manuscript.*

**We appreciate the referee comments on the importance of our results and thank the suggestions for further improving it.**

*Abstract Line 6-10: Too long statement. Break it into two.*

**Complied with. The sentence is modified as follows**

**"Concurrent measurements of the altitude profiles of cloud condensation nuclei (CCN) concentration, as a function of supersaturation (ranging from 0.2 to 1.0%), and aerosol optical properties (scattering and absorption coefficients) were carried out aboard an instrumented aircraft across the Indo-Gangetic Plain (IGP) just prior to the onset of the Indian summer monsoon (ISM) of 2016. The experiment was conducted under the aegis of the SWAAMI - RAWEX campaign. The measurements covered coastal, urban and arid environments."**

Line 15: "...followed by that in the west"

**Complied with**

Line 17: "...at all altitudes"

**Complied with**

Line 20: "The Great Indian Desert", also known as "The Thar Desert"

**Complied with**

Line 23-24: Due to washout of aerosols?

**The sentence is modified as "Wet scavenging associated with precipitation episodes seems to have reduced the CCN activation efficiency below cloud level"**

Page 4, line 5-10: Authors may cite following references related to the seasonality of aerosols (Jethva et al., 2005) and trends (Dey and Girolamo, 2011) over IGP.

Jethva, H., Satheesh, S. K., and Srinivasan, J. (2005), Seasonal variability of aerosols over the Indo Gangeticˇ basin, J. Geophys. Res., 110, D21204, doi:10.1029/2005JD005938.

Dey, S., and Di Girolamo, L. (2011), A decade of change in aerosol properties over the Indian subcontinent, Geophys. Res. Lett., 38, L14811, doi:10.1029/2011GL048153.

**Thanks for the suggestions. The mentioned references are included in the manuscript.**

Page 4, line 16: "...one of the best natural laboratories for investigating the complex nature of aerosols on clouds and precipitation."

**Complied with**

Page 5, line 5-7: The paragraph seems to be ending abruptly. Here, the author should mention concisely about the overall goals of the airborne experiments and objectives of the paper. Also, a brief writeup about how the analysis was conducted and what they were looking for would help the reader to familiarize with the overall content of the paper.

**Complied with. A brief write up as suggested is added in the manuscript as follows after Line 5,**

**"The campaign was planned to quantify the vertical distribution of total aerosols (CN), and CCN concentrations at different supersaturations and its spatial variation across the IGP, just prior to the onset of Indian summer monsoon, when different aerosol types are known to co-exist over this region. The data are analysed to understand the altitude distribution of CCN characteristics, its activation efficiency and its relationship with scattering and absorbing properties of aerosols and the variation of these from west to east across the IGP."**

*Page 5, line 12: "before the onset of ISM over central and northern India"*

**Complied with**

*Page 5, line 13-14: "also shown in Figure 2"*

**Complied with**

*Page 5, line 20: "…at the southern peninsular coast of Kerala state"*

**Complied with**

*Page 7, line 5: "All aircraft sorties…"*

**Complied with**

*Page 7, line 12-13: "Due to the unpressurized…"*

**Complied with**

*Page 7, line 5-10: Wouldn't be good if another sub-plot of the ratio of CCN-to-CN is added here?*

**Vertical profile of CCN activation efficiency (ratio of CCN to CN) is given in Figure 8 and discussed.**

*Page 13, line 13: "there seem to be a notable difference in the hygroscopicity of aerosols. . .". Author needs to be specific here though the hygroscopicity of aerosols is associated with aerosol type.*

**We did not see this line (underlined above) in the manuscript. However, we have pointed out in the manuscript that the changes in hygroscopicity along the vertical as well as across the IGP is also due to change in aerosol types.**

**This is mentioned in Page 17, Line-7 and Page 22, Line-12.**

*Page 15, Figure 4: The CN-CCN relationship over all three stations looks near-linear for the CCN range up to 3000-5000 cm-3, after which it becomes non-linear irrespective of the difference in aerosol type. Interesting.*

**Yes, we too agree. We have included this observation on page 14, line 14**

**"The linear association between CCN-CN for low to moderate CCN concentrations (up to ~ 4000 cm$^{-3}$) becomes non-linear for higher concentrations (CCN>5000 cm$^{-3}$). The CCN concentration tends to saturate at about 4000 cm$^{-3}$ (for 0.4% supersaturation), even though the CN concentration increases beyond 10000 cm$^{-3}$. Similar saturation of CCN concentration associated with large CN concentrations was reported by Roy et al., (2017) at ~ 2.2 km a.m.s.l. in Eastern Himalayas."**

*Page 16, Figure 5: If possible, please reverse the colors in the scale, i.e., blue for lowest altitude, red for the highest.*

**Complied with. The colour code of Figure 5 is modified as per the reviewer suggestion as shown below.**

[Figure]

**Figure 5: Association of CCN number concentration at 0.4% supersaturation with BC mass concentration over the east – BBR (circle), central – VNS (triangle), and west – JDR (square) IGP regions. Colour code indicates the altitude of observation**.

*Page 17, line 5: …represented by BBR, blue lines in Figure 6 (?). Also, is the flattening of CCN curve with SS for BBR an indication of aerosol type/size. Due to its proximity to the coast, BBR is likely influenced by coarse sea-salt particles against finer size aerosols over interior IGP. Throughout the discussion in this section, the author should cite corresponding aerosol studies supporting the link between aerosol type/size and CCN spectra.*

**Sorry for the typo error in the figure number, which is now corrected as 6. Additional references are added in the discussion as follows,**

**"Lower k values are reported more frequent for marine airmass compared to continental airmass (Twomey and Wojciechowski, 1969; Khain, 2009). The fine mode anthropogenic aerosols exhibit high k values, while hygroscopic and larger aerosols like seasalt have low k values (Hegg et al., 1991; Jefferson et al., 2010)."**

*Table 2. A concurrent geographical plot showing k values with colored circles as a symbol would be more effective and easier for a reader to grasp the reported values.*

**We found it difficult to convey the idea (vertical variation) by a spatial plot. As such, we included Table 2, which lists k values observed in the current study and reported by other investigators at different altitudes, even at the same location.**

*Page 28, line 15: slopes for BBR and VNS are comparable.*

**Complied with. Line 16 is modified as, "The highest slope is observed at the least anthropogenically impacted / dust dominated Western IGP, while the slope values are comparable over anthropogenically influenced East and central IGP."**

*Page 28, Figure 11: Maintain the same X-axis range for all three sub-plots.*

**Complied with. Figure 11 is modified and given below.**

[Figure]

**Figure 11: Association between the total scattering AI at 450 nm and CCN number concentration at 0.4% supersaturation for (a) eastern (BBR) (b) central (VNS) (c) western (JDR) IGP regions. The colour indicates the altitude of measurement. Dashed lines represent linear least-squares fit to the points for each region. Regression slopes and squared correlation coefficients are written in each panel.**

*Page 28, line 17-18: While this assumption generally holds, it would be interesting to plot CCN as a function of extinction aerosol index. Since the aircraft measurements delivered both scattering and absorption coefficients, it would be straightforward to create a similar plot using extinction AI.*

**It is a good suggestion and we have complied with thanks,**

A scatter between the Extinction aerosol index and CCN concentration at 0.4% supersaturation is generated and shown in Figure 12. If absorption contributed insignificantly to the extinction, then this plot would not differ significantly from Figure 11. However, it can be seen in Figure 11 that there is a significant reduction in the slope over western and Central IGP (JDR and VNS). This indicates the reduction in CCN activation due to absorbing aerosols, probably dust. However, there is no remarkable change in the slope over BBR, which might be due to the reduced concentration of dust (as most of it get removed as dust is advected across the IGP and also due to mixing of dust with other more hygroscopic aerosol species as it gets aged in the atmosphere). There is an increase in correlation coefficient over east IGP when we consider aerosol absorption also, which might be indicative of contribution of these aerosols to CCN activation; probably due to co-emitted or co-existing soluble inorganic particles. Figure 12 and the above discussions are included in the revised manuscript.

[Figure]

**Figure12: Association between the Extinction AI at 450 nm and CCN number concentration at 0.4% supersaturation for (a) eastern (BBR) (b) central (VNS) (c) western (JDR) IGP regions. The colour indicates the altitude of measurement. The solid lines represent linear least-squares fit to the points for each region. Regression slopes and squared correlation coefficients are written in each panel.**

*Page 29, line 23-24: The relationship between CCN and aerosol properties further implied the use of satellite-retrieved AOD products in the region, which are now matured and fairly accurate, and model-generated aerosol profile aided by ground (MPLNET at Kanpur)/space lidar (CALIOP), in predicting CCN. This should be mentioned here and also in the conclusion.*

**Complied with.**

*Also, did author check the relationship between CCN efficiency and aerosol scattering/extinction properties, if any? It is worth to perform such analysis.*

**Yes, we have checked the relationship and is shown in Figure ii. From the Figure it can be seen that there is no clear association between CCN activation efficiency and aerosol scattering coefficient. The CCN efficiency is an intrinsic property which quantify the water affinity of the given aerosol system at a particular supersaturation, while aerosol scattering coefficient is an extrinsic property, which depends on the abundance of the aerosols. For eg. higher the aerosol loading may result in larger scattering coefficient, but need not increase the CCN efficiency. Liu and Li, (2014) have suggested that the Scattering Aerosol Index, which includes both aerosol abundance and size information is a better proxy to relate with CCN.**

[Figure]

**Figure ii: Association of CCN efficiency with scattering coefficient over BBR (circle), VNS (triangle), and JDR (square).**

*Page 30, line 5: "…towards characterizing/understanding the ACI…"*

**Complied with.**

**References:**

**Khain, A.P., 2009. Notes on state-of-the-art investigations of aerosol effects on precipitation: a critical review. Environ. Res. Lett. 4 (1), 015004. http://dx.doi.org/ 10.1088/1748e9326/4/1/015004.**

**Twomey, S.,Wojciechowski, T.A., 1969. Observations of the geographical variation of cloud nuclei. J. Atmos. Sci. 26 (4), 648e651**

**General comments:**

*The manuscript presents the altitude variations of CN and CCN characteristics in the troposphere below 3 km based on systematic aircraft based observations carried out over the western, central and eastern IGP during June 2016, just before the onset of Indian summer monsoon. Altitude variations of CCN activation efficiency, CCN spectra and their similarities and differences over the three regions are investigated, in the light of the potentially different aerosol sources. One of the most striking features observed is the high CCN activation efficiency over the dust- dominated western IGP. The topic of research is highly relevant and current and the results presented here are of very high importance for a wide range of scientific community (including aerosols, aerosol-cloud interaction and climate modelling studies).*

*The manuscript is well written and the conclusions are well supported by observations.*

**We appreciate the summary evaluation and the positive comments of the reviewer.**

*I have a few minor comments/suggestions; addition/modification of a few sentences will be sufficient to address all these comments. Considering the very high scientific importance of the results presented, which are of interest to a wide scientific community, I recommend that the manuscript may be accepted for publication in ACP after minor revision.*

**We have gratefully complied with the suggestions and taken care of the comments in the revised manuscript**

*Page-22, Paragraph-2 & Fig.8(b): This information is redundant as it can be inferred from Figs. 6, 7, and 8(a) and the discussions on them. However, if the authors want to still keep it, please spell out the following: Lines 18-19: what is the range of 'low CCN efficiency' and 'low k values' referred here? I think this result is not very evident in Fig.8(b).*

**The inverse relationship between the CCN activation efficiency and k value, seen in Figure 8(b) is significant for the characterisation of the CCN activation of a given aerosol system. CCN efficiency is determined from two independent instruments (CCN counter and CPC), while k is estimated from the response of the ambient aerosols to a given range of supersaturations. The inverse relationship will be different for different aerosol systems even for a given supersaturations. Thus, Figure 8 (b) demonstrates the variation of k with CCN efficiency at 0.4 % ss across the IGP.**

**At VNS, even when the CCN efficiency is very low (<20 %), the corresponding k values are not high (not steep), as expected. For 19.9 and 17.3 % CCN efficiency, the k values were 0.43 and 0.46, respectively. These points were observed within**

**PBL and are mentioned in the manuscript. For clarity the Lines 18-19 are modified as**

**"However, over central IGP, very low CCN efficiency (<20 %) were observed with low k values (~ 0.4), which is not in-line with the general inverse relationship. These cases were observed within the PBL, indicating a CCN-inactive aerosol system even at high (>0.8 %) supersaturations."**

*Page-23, Lines 17-18: "The airmass traversing through the polluted-continental region is responsible for the lowering of CCN activation efficiency at the free troposphere heights over the east IGP". Note that, among the three regions considered here, the CCN efficiency is highest at all altitudes over BBR (e.g., Fig.8a). The above statement can be true if the CCN activation efficiency is found to be higher when the tropospheric airmass transport over BBR is from the east compared to those from the west. You may clarify how this conclusion was arrived at? Please delete the sentence if it cannot be explained unambiguously.*

**The CCN activation efficiency was highest over BBR at all altitudes compared to other regions of IGP. But there is a decrease in the efficiency at the highest altitude (3 km). This is due to the continental airmass reaching at that level, while marine airmass prevailed at lower altitudes. This is clearly seen in Figure S1. The blue line in the Figure indicates the mean (with standard deviation) airmass back trajectories reaching at 500 m above BBR, which is having a considerable marine influence. The mean (with standard deviation) airmass back trajectories (red) reaching at 3000 m above BBR is completely continental. The distinctiveness in the airmass history at higher altitudes are also causing the scatter in CCN-CN association.**

**This clarification is included in the manuscript as, "The back trajectory analysis of airmass reaching at 500 m and 3000 m over BBR (figure not included) clearly showed that the particles reaching 3000 m have pure continental history of passing across the IGP from the arid regions of western India and West Asia, whereas those reaching at 500 m pass over oceanic region of Bay of Bengal before arriving at the location. This distinctiveness in the airmass history at higher altitudes are also causing the scatter in CCN-CN association as seen in Figure 4."**

[Figure]

**Figure S1: Mean airmass back trajectories reaching at 500 m (blue) and 3000 m (red) above BBR.**

*Page-24, second paragraph: This is a very interesting and, perhaps, the most important finding from this study. It has major implications in ACI.*

**Complied with**

*Page-25, Line 18: "... while the air is deprived of moisture;...". Note that 'k' is related to the property of aerosols (size distribution and water affinity, as stated in the manuscript) and is measured by systematically changing the supersaturation inside the instrument. Then, how "depriving of moisture" in the atmosphere will result in high value of 'k'?*

**Thanks for pointing out this issue and we agree. As rightly pointed out atmospheric moisture has no role in estimated k value. The phrase, "while the air of deprived of moisture" is removed in Line 18.**

*Page-26, Lines 8-9: "... and at VNS a rainfall of 20 mm occurred on the evening of 7th June". What is relevance of this statement here? Dependence of any aerosol/CCN parameter on rainfall at VNS is not presented in this manuscript.*

**Since a distinct airmass was predominantly influencing the CCN vertical profile on 8 June, we were not able to delineate the effect of previous evening rainfall. To reduce complexity, we are removing the rainfall information (at VNS) statement in Line 8-9.**

*Page-26, Line-15: "After the rainfall, a reduction (<10%) is seen in the CCN efficiency over BBR, ...". This is true for the height range below 2000 m, while the opposite is the case for 2500-3000 m (Fig.10). Is it because of the difference between in-cloud and below-cloud processes that remove/shift the size*

*distribution? Also see the comment below. If it cannot be explained based on the present set of observations, please include a line on the differences (CCN efficiency) observed in the altitude range of 2500-3000 m.*

**From the current observations we could not find the exact reasons for the altitude-wise difference in CCN efficiency due to rainfall. Hence, we are adding the information in the manuscript as,**

**"Even though the CCN efficiency is found to be slightly reduced below 2 km, the CCN activation efficiency is found to be higher above 2 km compared to that of observations before the rainfall"**

*Page-27, Lines 14-17: How this process (more efficient removal of CCN by in-cloud scavenging) can enhance the CCN efficiency after rainfall (height range of 2500-3000 m; Fig.10)?*

**Agree with the reviewer and hence removing the confusing Lines 14-15. The explanation is modified in the manuscript as, "The difference in CCN activation efficiency at different altitude levels before and after rainfall reinstates the difference in the aerosol types at different altitudes. One of the possibilities for the observed CCN efficiency is that the rainfall has removed coarser and hygroscopic particles by wet scavenging, resulting in the reduction of the CCN activation efficiency below 2 km. Cloud processing broadening the aerosol distribution as reported by Flossmann et al., (1987) may be enhancing the CCN activation efficiency above 2 km. However, the effect of cloud formation and further rainfall on CCN characteristics needs further investigation."**

*Page-28: Lines 8-12: Modify this sentence (it is not very clear; contains 'because' twice).*

**Complied with. Lines 14-17 are modified as,**

**"Examining Figure 11 along with the CN profile shown in Figure 3(a), it can be seen that the higher slope (21.8) at JDR is due to the large size dust particles there, even though the CN concentrations at JDR and BBR are comparable, except at the lowest altitude. The coarse size distribution would lead to smaller scattering Angstrom exponent resulting in low scattering AI values. It is interesting to note that scattering AI values at JDR are low, though the scattering coefficient values are higher than BBR (Vaishya et al., 2018)"**

*Page-28, Lines 11-12: "... implying that the total scattering coefficient would have to be of comparable magnitude". Why this guess? You already have the scattering coefficient measurements available (used for estimating AI). Did I miss something?*

**Sorry for the lack of clarity. The sentences (Line 14-17) are modified according to the previous comment and given above.**

*Page-30 (Conclusions), Lines 17-18: "High CCN activation efficiency ... dust dominated western IGP". This is a very interesting and important result. A statement on its implication will be highly useful.*

**Complied with. "This high CCN activation efficiency of dust aerosols can modify the cloud microphysics over the region, hence affecting the precipitation pattern as well as the regional radiation balance".**

**Other suggestions:**

*Page-4 Line 20: Keep proper reference format*

**Complied with**

*Page-5 Line 14: Change : "Synoptic wind ..." as "Monthly mean synoptic wind ..."*

**Complied with**

*Page-11, Line 1: expand "ss" (first time usage of ss for supersaturation)*

**Changed to supersaturation in the manuscript.**

*Figure 4: If you have sufficient ancillary data required, it would be interesting to know why there are major deviations from the general trend on (i) Day-1 at BBR and (ii) Day-4 at VNS. Is this the effect of rain or change in airmass trajectory?*

**As rightly pointed out, the day-1 at BBR experienced a distinct stagnant continental airmass reaching especially at 3 km above ground level. This is mentioned in Lines 6-9 in page 14. But over VNS, except June 8, all the airmass had predominant continental influence. However, with the current dataset, it is difficult to exactly attribute the reasons.**

**References:**

**Flossmann, A.I., Pruppacher, H.R. and Topalian, J.H., 1987. A theoretical study of the wet removal of atmospheric pollutants. Part II: The uptake and redistribution of (NH4) 2SO4 particles and SO2 gas simultaneously scavenged by growing cloud drops. *Journal of the atmospheric sciences, 44*(20), pp.2912-2923.**